# Recent Advancements in the Control of Cat Fleas

**DOI:** 10.3390/insects11100668

**Published:** 2020-09-29

**Authors:** Michael K. Rust

**Affiliations:** Department of Entomology, University of California, Riverside, CA 92521, USA; michael.rust@ucr.edu

**Keywords:** *Ctenocephalides felis felis*, isoxazolines, essential oils, insecticide resistance

## Abstract

**Simple Summary:**

The cat flea *Ctenocephalides felis felis* is the most important pest of domesticated cats and dogs worldwide. This review covers the recent advancements in the control of cat fleas. Over the years, there has been an interest in using ecologically friendly approaches to control fleas. To date, no biological, natural, or cultural means have been discovered that mitigate flea infestations. The recent registration of novel topical and oral therapies promises a new revolution in the control of fleas and ticks and the diseases associated with them.

**Abstract:**

With the advent of imidacloprid and fipronil spot-on treatments and the oral ingestion of lufenuron, the strategies and methods to control cat fleas dramatically changed during the last 25 years. New innovations and new chemistries have highlighted this progress. Control strategies are no longer based on the tripartite approach of treating the pet, the indoor environment, and outdoors. The ability of modern therapies to break the cat flea life cycle and prevent reproduction has allowed for the stand-alone treatments that are applied or given to the pet. In doing so, we have not only controlled the cat flea, but we have prevented or reduced the impact of many of the diseases associated with ectoparasites and endoparasites of cats and dogs. This review provides an update of newer and non-conventional approaches to control cat fleas.

## 1. Introduction

One of the most important pests of domestic cats and dogs is the cat flea, *Ctenocephalides felis felis* (Bouché). Several reviews concerning the biology and control of cat fleas and new therapeutics have been published [1,2,3,4,5,6]. However, it has been over 20 years since the topic of alternative measures to control cat fleas has been reviewed [7]. Numerous advancements resulting from the development of new spot-on and oral therapeutics have occurred since then. These advancements have changed our thinking and approach to managing fleas on pets and in the indoor environment.

Even though there has been an increased awareness in so-called “green pest management” in the urban environment, limited progress in controlling ectoparasites with natural products or biological agents has been made. With the advent of the spot-on treatments of fipronil and imidacloprid and the systemic use of lufenuron in the mid-1990s, a paradigm shift in our thinking regarding flea control occurred. In the last decade, treatment of the pet with many of the new therapies has made it possible to interrupt the life cycle of the cat flea and prevent reproduction, thereby eliminating the need to spray interior and exterior environments with insecticides to effectuate control.

The objectives of this review are to update the status of flea control and to provide additional insights into strategies to control cat fleas, especially new therapies and their impacts on other arthropod ectoparasites, and problems associated with cat fleas.

## 2. Biological Control

Even though some studies with bacteria and fungi have demonstrated activity against cat fleas, the research into biological control of cat fleas has been limited [6]. In a recent study, adult cat fleas were exposed to fungal spores, *Beauveria bassiana* (isolates 4849, 2MG), exposed to different lighting conditions to stimulate conidiation. Spores produced under a red LED light and fluorescent lighting produced the fastest mortality, killing 100% of adult cat fleas within 36 h [8]. It remains uncertain how this might be exploited to control fleas.

## 3. Vaccines

The search for a vaccine to protect cats and dogs from cat fleas has persisted over the past four decades. Vaccines have the advantages of not contaminating the environment, avoiding the development of insecticide resistance, targeting a broad, but selective range of vector species, and reducing vector competence [9]. Despite these benefits, developing a vaccine has been problematic. Obstacles to developing a vaccine include the lack of natural immunity of cats and dogs to flea infestations, difficulties in obtaining large quantities of flea extracts, and the presence of a serine proteinase in the flea midgut [10]. To date, there is no vaccine available against insect ectoparasites.

The identification of 97 distinct expressed sequence tags that encoded proteins of the cat flea hindgut and Malpighian tubules may provide molecular targets for flea control strategies in the future [11]. Vaccination of cats with recombinant antigens resulted in an antibody response. Efficacy was defined by determining flea mortality and fertility, oviposition, and viability of the flea eggs. It reduced cat flea egg hatchability and fertility, resulting in a 32–46% efficacy [12].

To relieve the effects of flea allergy dermatitis (FAD) in cats, a co-immunization study using DNA encoding flea saliva antigens and proteins suppressed T cell reactions was conducted. It ameliorated many of the clinical symptoms of FAD in cats and showed utility in a clinical study [13].

Despite these findings, it seems like a vaccine to control fleas is a long way in the future, but an anti-allergic vaccine may be available sooner.

## 4. Botanical Based Compounds

In the past decade, there has been an increased interest in the use of essential oils (EOs) to control insects of urban and veterinary importance [6,14]. Several products containing *d*-limonene were registered and tested against cat fleas [1,2]. Researchers continue to identify natural products and EOs that are toxic to fleas. Deposits of carvacrol and nootkatone applied inside glass vials were toxic to adult oriental rat fleas, *Xenopsylla cheopis* [15]. Extracts from incense cedar, Port Orford Cedar, and western juniper were toxic to adult *X. cheopis* when applied to the inner surfaces of glass vials [16]. Extracts of California pepper tree (*Schinus molle*) applied to filter paper were toxic to adult *C. f. felis* but failed to kill flea eggs [17]. Adult fleas exposed to filter paper strips treated with EOs from *Ocimum gratissimum* (clove basil) and *Cinnamomum* spp. (cinnamon) were killed. Filter papers treated with clove oil at 25 μg/cm^2^ killed 100% of adult fleas at 24 h. The EOs were also active against larvae and eggs [18]. The only study involving pets and EOs reports on various infusions or preparations of six plants, mugwort, lemon, juniper, lavender, lemon balm, and cedar, against fleas, but the paper lacks experimental validation [19].

Plant-derived flea products have been reported to have some adverse effects, especially when applied to cats. Topical applications of tea tree oil (Australian tea tree *Melaleuca alternifolia*) to cats and dogs resulted in depression, weakness, incoordination, and muscle tremors [20]. In another case, *Melaleuca* oil applied to three cats resulted in death and severe reactions and toxicosis [21]. About 5–9% of the cats treated with spot-on products that included EOs such as peppermint oil, cinnamon oil, lemongrass oil, and clove oil experienced conditions, such as higher agitation, hypersalivation, seizures, and lethargy [22]. Addie et al. [23] recommend that some EOs should only be used after consultation with a veterinarian.

Several studies have examined EOs as repellents. The European Medicine Agency [24] defines a repellent effect as “a product with a repellent effect will cause the parasite to avoid contact with a treated animal completely and/or to leave a host.” A choice bioassay on filter papers treated with EOs provides a rapid laboratory determination of repellency to adult fleas. Essential oil from *Cinnamomum osmophloeum* (leaves), *Taiwania cryptomerioides* (heartwood) and *Plectranthus amboinicus* (leaves) exhibits repellent activity against cat fleas in a dose dependent manner [25]. Extracts of the seeds of monk’s pepper (*Vitex agnus castus*) repelled cat fleas for about 6 h [26]. It is unclear how these EOs might be used in a control program to deter flea infestations, especially considering their potential negative impacts on cats.

## 5. Chemical Treatments

Several reviews have covered the development and efficacy of various therapies to control cat fleas [1,2,6,27]. In recent years, the research and development focus has been on treating the pet, rapidly killing adult fleas, and preventing the flea life cycle from maintaining itself. 

### 5.1. Aminoglycosides

Spinosad is an insecticidal compound derived from natural products from the fermentation of the bacterium *Saccharopolyspora spinosa*. Given orally to dogs, it provides >95% kill of *C. f. felis* for several months. Spinetoram, an analogue of spinosad, is even more active than spinosad [6]. A laboratory study with dogs indicated that 30–60 mg/kg of spinetoram provided excellent kill of adult fleas for 2–3 months [28]. In a laboratory and field study conducted in Europe, topical application of spinetoram to cats provided 99.1% reductions of *C. f. felis* at day 60 whereas fipronil + methoprene provided a 92.3% reduction [29]. Topical applications of spinetoram provided 96% reductions through day 37, providing about the same speed of activity against adult fleas as imidacloprid and fipronil + methoprene [30].

### 5.2. Insect Growth Regulators

The use of insect growth regulators (IGRs), such as methoprene and pyriproxyfen to inhibit egg hatching and larval development in the environment, and oral applications of lufenuron to interfere with egg viability, have played important roles in developing our understanding of successful cat flea pest management. Combination treatments with adulticides and IGRs interrupted the flea life cycle and controlled fleas inside residences [6,31,32].Other IGRs, such as chlorfluazuron and dicyclanil (0.78 and 0.3 ppm lethal concentration [LC_95_] in larval rearing medium), are as active as methoprene and pyriproxyfen against cat fleas, and may be promising candidates [33]. Interestingly, pyriproxyfen synergized the activity of methoprene against larval cat fleas with a combination of pyriproxyfen:methoprene (10:1) being twice as active (LC_50_ 0.20 ppm in larval rearing medium) as either methoprene or pyriproxyfen alone [34]. The synergism permitted lower concentrations of each IGR to be applied and still be effective.

The combination of imidacloprid and methoprene and imidacloprid and pyriproxyfen were synergistic against larval fleas [35]. Some combinations of fipronil and methoprene or fipronil and pyriproxyfen were synergistic, but in some cases, they were antagonistic. Synergistic combinations allow the concentrations of the adulticide and IGR to be reduced, and still provide the same activity as higher concentrations of the adulticides alone. In a laboratory study, cats topically treated with fipronil + methoprene + pyriproxyfen were allowed to contact carpets. Pieces of carpet were removed and provisioned with flea rearing medium and cat flea eggs. The plugs of treated carpets prevented 86–98% egg development over a 15-week period, effectively interrupting the life cycle [36].

In 2013, a spot-on to treat dogs containing fipronil and novaluron was registered [37]. To date there are no published results concerning its efficacy.

### 5.3. Isoxazolines

An exciting new class of insecticides, the isoxazolines, was registered in 2014 and emerged on the ectoparasiticide market in 2015 [27]. Isoxazolines are potent inhibitors of γ-aminobutyric acid (GABA)-gated channels and affect l-glutamate-gated chloride channels to a lesser extent [38,39,40,41]. They are second-generation non-competitive antagonists of GABA receptors. Isoxazolines circumvent resistance by attacking new binding regions on the chloride channels [39]. No cross-resistance exists with other non-competitive antagonists, such as fipronil or the macrocyclic lactones such as abamectin and emamectin benzoate [42,43]. Enough differences between insect and mammalian receptors occur to make them selective to insects, and Acari, and excellent candidates as ectoparasiticides. The toxic effects may also extend to Crustacea, as lotilaner was shown to be a powerful antagonist GABA chloride channel of sea lice [38]. Laboratory studies and field trials of new isoxazoline compounds, and combination products published prior to 2017 have been reviewed by Rust [6]. To date afoxolaner, fluralaner, lotilaner, and sarolaner are commercialized.

In addition to killing adult cat fleas, the isoxazolines also kill mites, ticks, lice, triatomine bugs, mosquitoes, biting flies, and sea lice. Some endoparasites also appear to be affected. This remarkable breadth of activity against Insecta, Arachnida, and Crustacea will revolutionize the control of endo- and ectoparasites on cats and dogs. In a study in Sicily and southern Italy, a topical treatment of fipronil + permethrin and an oral dose of afoxolaner + milbemycin oxime was provided to each dog. In addition to killing fleas and ticks, the combination treatment provided a broad spectrum of protection against *Anaplasma* spp., *Borrelia* spp., *Ehrlichia* spp., and dog heartworm [44].

The Project Lake Survey of veterinarians and dog owners regarding their experiences with isoxazoline products indicated that 48.2% of the 2751 respondents had used them [45]. Of the 1768 positive respondents, 66.6% indicated that the dogs had responded with adverse events to the treatment ranging from hair loss to death. As the authors indicate, there are a number of limitations in evaluating voluntary survey data. Palmieri et al. [45] write, “However, FDA (Food and Drug Administration) and EMA (European Medicine Agency) AE (adverse event) reports and Project Lake Survey evidence consistently demonstrate in three separate, distinct data sets that the neurotoxicity is not arthropod-specific, and that post-marketing serious AE are much higher than in the IND (Investigational New Drug) submission studies.” Clearly, additional investigations and surveys are warranted.

#### 5.3.1. Afoxolaner

Afoxolaner first appeared on the market in 2014 as an oral chewable dose for dogs. A chronology of the development of afoxolaner for dogs is provided by Letendre et al. [46]. In addition, afoxolaner also was topically active against fleas and ticks. In a laboratory study with cats, a single oral dose of afoxolaner provided 100% kill of adult fleas within 48 h and 99.5 and 95% reductions for 42 and 63 days, respectively [47]. In a laboratory study, the number of fleas on dogs orally dosed with afoxolaner was reduced by 100% for at least 22 days [48]. An oral dose of afoxolaner + milbemycin oxime to naturally infested dogs provided ≥96.1% reductions of fleas at day 30 [49]. In a field trial with 37 naturally infested dogs in Tampa FL, USA, dogs were provided an oral dose of afoxolaner. Within 7 days, there was a 93% reduction in the number of fleas counted on the dogs and a 99.9% reduction at 30 days. There was a 100% reduction in the number of adult fleas emerging within the residences at 60 days [50].

In a laboratory study, there was 100% kill of the tick *Haemaphysalis elliptica* on dogs treated with afoxolaner, and it protected them from contracting *Babesia rossi* [51]. In a laboratory study with dogs, fipronil + permethrin spot-on provided faster knockdown and greater repellency of ticks that did an oral application of afoxolaner. Both treatments provided >90% anti-attachment of *Rhipicephalus sanguineus* for at least 14 days [48]. An oral dose of afoxolaner + milbemycin oxime provided ≥94.4% reductions of ticks on naturally infested dogs at day 30 [49].

#### 5.3.2. Fluralaner

In a simulated home environment, where natural environmental infestations could reinfest pets, a single topical application of fluralaner provided nearly 100% reductions of fleas on cats and dogs for 12 weeks [52]. In a laboratory study with cats infested with a field isolate of *C. f. felis* not controlled by fipronil, a spot-on application of fluralaner + moxidectin provided nearly 100% control for the entire 93-day study, whereas a topical application of fipronil + methoprene provided from 65.6 to 30.6% reductions over the same period [53]. In a laboratory study with short haired cats, a single topical application of fluralaner or three topical applications monthly of selamectin + sarolaner provided >94.6% kill of adult cat fleas for 90 days [54].

Topical fluralaner applied to naturally infested cats from 18 different veterinary clinics across 11 USA states provided 99.0% reduction of *C. f. felis* at week 12. Topical applications of fipronil + methoprene provided a 75.4% reduction [55]. In a similar study with dogs, topically applied fluralaner and fipronil + methoprene provided 99.9 and 93.0% reductions of cat fleas at week 12 [56]. A single topical application of fluralaner to naturally infested cats provided 100% reductions of *C. f. felis* for up to 84 days [57].

In a large study with 707 cats from 332 households in Germany and Spain, a single spot-on application of fluralaner + moxidectin on cats provided 97 and 98% reductions in ticks and fleas, respectively, throughout the 12-week study. A topical fipronil applied for three consecutive months provided 74.5% reductions at 12 weeks [58].

#### 5.3.3. Lotilaner

An oral dose of lotilaner given to dogs provided 100% kill of adult fleas within 24 h. There was a 98.5% reduction in the number of eggs laid at 24 h and no eggs were laid any period afterwards. When the dogs were challenged with adult fleas, there was a 100% kill for 30 days [59]. An oral dose of lotilaner to cats with an existing *C. f. felis* infestation provided 100% kill within 24 h. In a second study, 97.4% of the fleas were killed within 8 h. When cats were challenged with adult cat fleas, lotilaner provided 98.6% kill for at least 35 days [60].

Several studies have been conducted to determine the speed at which fleas are killed by lotilaner. In a laboratory study with dogs with existing infestations of *C. f. felis*, lotilaner provided 64 and 99.6% reductions in the number of fleas counted at 2 and 8 h, respectively [61]. An oral dose of lotilaner to dogs provided 89.9% kill of existing cat flea infestations at 4 h and 100% kill at 12 h. When treated, dogs were challenged with adult fleas for up to 35 days, there was >97% kill at 4 h [62].

Client-owned dogs from 10 veterinary clinics in the USA were dosed with lotilaner or afoxolaner. Lotilaner reduced the number of fleas by 99.3, 99.9, and 100% at 30, 60, and 90 days, respectively. Afoxolaner provided 98.3, 99.8, and 99.8% reductions at 30, 60, and 90 days, respectively. On day 90, all the dogs dosed with lotilaner were flea free and 93% of dogs were flea free when treated with afoxolaner [63].

In a laboratory study, an oral dose of lotilaner to dogs provide 100% kill of *Ixodes ricinus* within 8 h. Treatment provided protection for 35 days [64]. Similarly, in laboratory studies with four common species of ticks, an oral dose of lotilaner provided greater than 98% efficacy against *Dermacentor variabilis*, *R. sanguineus*, *Amblyomma americanum,* and *Ixodes scapularis* for at least 4 weeks [65]. Similarly, a single oral dose of lotilaner to dogs provided >98% efficacy against *I. ricinus*, *R. sanguineus*, and *Dermacentor reticulatus* for at least 35 days [66]. In a laboratory study, a single oral dose of lotilaner provided >97% reduction in the number of *Haemaphysalis longicornis* attached to dogs at 48 h [67].

#### 5.3.4. Sarolaner

The development of sarolaner and supporting studies are discussed by Woods and McTier [68]. Sarolaner was about 10 times more toxic to *C. f. felis* than afoxolaner or fluralaner in membrane feeding studies. Oral doses of sarolaner provided 100% reduction in the number of fleas retrieved from dogs for up to 35 days. The activity of sarolaner was not negatively affected by the dieldrin resistant mutation at CfRDL-S285 channel [68,69]. In a laboratory study with dogs, topical applications of sarolaner to dogs provided 100 and 87% kill of fleas at day 1 and 28, respectively, after a six-hour challenge. The treatment with fipronil + methoprene + pyriproxyfen provided 88.8 and 83% reductions of fleas at days 1 and 28, respectively [70].

In a field study in USA, 479 dogs in 293 households were given oral doses of sarolaner or spinosad monthly for 3 consecutive months. At day 90, sarolaner and spinosad provided 99.9 and 99.8% reduction in the number of fleas on dogs, respectively [71]. In a field study conducted in west Central Florida, 26 dogs were orally dosed with sarolaner or a spinosad chewable. Both treatments provided >99% reduction of fleas for at least 60 days. The number of fleas trapped in the structures was reduced by 100% in sarolaner and 99.8% in the spinosad treated dogs [72].

The combination of selamectin + sarolaner topically applied to cats in Europe provided a 99.4% reduction in flea counts at day 90. A comparative treatment of imidacloprid + moxidectin provided a 96.3% reduction in the number of cat fleas [73]. In another study, topical application of selamectin + sarolaner provided a 99.8% reduction of *C. f. felis* at day 90. Clinical signs of flea allergy dermatitis (FAD) were reduced in 86.7 to 100% of the cats. A topical treatment of imidacloprid + moxidectin provided 95.5% reductions in the number of fleas counted at day 90 and 66.7 to 100% reduction in clinical signs of FAD [74]. In a field study in Japan, 67 cats were topically treated with selamectin + sarolaner or fipronil + methoprene. The selamectin + sarolaner provided 99.5 and 99.9% reduction in numbers of fleas on the cats at days 14 and 30, respectively. Fipronil + methoprene provided 97.6 and 98.6% at days 14 and 30, respectively [75]. In Australia, 104 cats were enrolled in clinical studies. A topical application of selamectin + sarolaner for 3 consecutive months provided 98, 100, and 100% control of cat fleas at days 30, 60, and 90, respectively [76].

In a laboratory study with dogs, an oral dose of sarolaner provided 86.2 and 96.5% reductions in *Ixodes holocyclus* at 8 and 12 h, respectively whereas an oral dose of afoxolaner provided 21.3 and 85.0% reductions at 8 and 12 h, respectively. When treated dogs were challenged at day 35, sarolaner and afoxolaner provided 65.2 and 21.0% efficacy at 12 h, respectively [77]. Oral doses of sarolaner, moxidectin, and pyrantel to laboratory dogs provided 99.7% reductions in *C. f. felis* for at least 35 days with no eggs being laid during the 35 days. The treatment began killing fleas within 4 h and all the fleas were dead at 8 h. When moxidectin and pyrantel were applied to dogs, they had no effect on fleas [78].

A clinical field trial of 150 dogs dosed with sarolaner + moxidectin + pyrantel provided a 99.0% reduction in cat flea counts at day 30 and a 99.7% reduction at day 60. Clinical signs of FAD declined from 45.7 to 6.9%. Similarly, in a field study with dogs in Europe and the USA, an oral dose of sarolaner + moxidectin + pyrantel provided ≥97.9% reductions in flea counts at day 30 [79]. In a large multi-location study in the USA, oral doses of sarolaner + moxidectin + pyrantel provided 99.0 and 99.7% at days 30 and 60, respectively [80].

In laboratory studies on dogs, an oral dose of sarolaner + moxidectin + pyrantel provided 98.9% kill of existing infestations of the five most common ticks in the USA. At day 28, >88% kill was achieved 48 h after exposing the ticks to the treated dogs [81]. In a laboratory test, an oral dose of sarolaner + moxidectin + pyrantel provided 100% efficacy of existing infestations of African yellow dog ticks, *H. elliptica*, on dogs and weekly re-infestations for 35 days [82]. Similarly, sarolaner + moxidectin + pyrantel provided 100% efficacy for 21 days against *H. longicornis* and ≥97.4% efficacy for 35 days [83]. In another study, sarolaner + moxidectin + pyrantel provided 99.4% kill of black legged tick *I. scapularis* at 24 h and provided a 94.2% reduction of ticks for at least 28 days [84].

In a field study in Japan, 67 cats were topically treated with selamectin + sarolaner or fipronil + methoprene. Selamectin + sarolaner provided 97.5 and 97.7% reductions of the number of the tick *H. longicornis* at days 14 and 30, respectively. Fipronil + methoprene provided 91.5 and 93.4% reduction in ticks at days 14 and 30 [75]. In a field study with naturally infested dogs, and an oral dose of sarolaner + moxidectin + pyrantel provided ≥94.8% reduction of ticks at 30 days. An oral dose of afoxolaner + milbemycin oxime provided ≥94.4% reduction in the number of ticks at day 30. The ticks included in the test were *I. ricinus*, *Ixodes hexagonus*, *R. sanguineus*, and *D. reticulatus* [85].

#### 5.3.5. Additional Uses

The label directions and control claims for each of the isoxazoline insecticides registered vary. In addition to controlling fleas and ticks, isoxazolines in combination with endoparasiticides (i.e., macrocyclic lactones) allow the broader spectrum to intestinal nematodes and heartworm.

In a recent review, the use of isoxazolines to control *Demodex* mites in dogs was concluded to be effective, with few adverse side effects [86]. Oral doses of afoxolaner or afoxolaner + milbemycin oxime to dogs provided 98.1% reduction in the number of *Demodex* mites at day 84. At day 84, 62.5% of the dogs were considered mite free. Skin lesions and pruritus were significantly reduced in the treated dogs [87]. In another study, a single oral dose of afoxolaner + milbemycin oxime to dogs provided >95% reduction in the number of *Demodex* mites by day 28. The most significant decreases in mites and lesions occurred during the first 7 days [88]. In a study of healthy dogs with populations of *Demodex* mites, cutaneous populations of *Demodex* over the 90-day period were not affected by a treatment of fluralaner or afoxolaner [89]. It was suggested that isoxazoline treatments may not completely eliminate *Demodex* mites, but only return them to a more natural population level.

An oral dose of afoxolaner to dogs provided 99.4% reduction of ear mites *Otodectes cynotis* at day 28 [90]. Topical applications of fluralaner to cats provided 100% control of the ear mite by day 7 and provided protection for at least 84 days [57]. A single application of selamectin + sarolaner provided 94.4% reduction of *O. cynotis* within 30 days compared with a 72% reduction when treated with imidacloprid + moxidectin [74]. An oral dose of afoxolaner to cats naturally infested with ear mites provided 100% efficacy for at least 35 days [91].

Oral doses of afoxolaner or afoxolaner + milbemycin oxime provided >98% reduction in the number of *Sarcoptes* mites from skin scrapings from dogs for 2 months [92].

An oral dose of afoxolaner or fluralaner to dogs provided 100% kill of the bug *Triatoma infestans* (a principal vector of Chagas disease) for 51 days. Less blood was consumed by bugs feeding on dogs treated with afoxolaner than fluralaner for some unknown reason [93].

Yellow-fever mosquitoes, *Aedes aegypti*, were allowed to feed on dogs dosed with afoxolaner. Mosquitoes readily fed on the dogs indicating that the treatment was not repellent. At 24 h after feeding, 98 and 75.3% of the female mosquitoes were killed when exposed to dogs on day 2 and 29 after the dogs were dosed, respectively [94].

The use of systemic insecticides to control phlebotomine fly vectors has been reviewed by Gomez and Picado [95]. The flies are responsible for transmission of zoonotic visceral leishmaniasis. An oral dose of fluralaner to dogs provided 60–80% mortality of phlebotomine flies for 30 days, but moxidectin, spinosad, and afoxolaner did not increase mortality of the flies [96]. In a laboratory study, dogs dosed with afoxolaner provided 100, 95.9 and 75.5% mortality of *Phlebotomus perniciosus* within 48 h after feeding [97]. Flies fed on both treated and untreated dogs and the afoxolaner was not repellent.

An indirect effect of treating pets with isoxazolines is the prevention of dog tapeworm transmission by infected adult fleas. An oral dose of afoxolaner + milbemycin oxime provided indirect protection to dogs from ingesting adult cat fleas infected with cysticercoid larvae of *Dipylidium caninum* [98]. Topical or oral fluralaner treatments prevented dogs from acquiring dog tapeworm [99].

An oral dose of sarolaner + moxidectin + pyrantel or afoxolaner + milbemycin oxime to dogs naturally infected with ascarid nematodes and hookworms provided >99% reduction of fecal egg counts [79]. A topical dose of selamectin + sarolaner prevented the development of feline heartworm *Dirofilaria immitis* in cats [100].

Dogs naturally infested with canine screwworm myiasis were treated with five different insecticides or drugs. Nitenpyram killed all fly larvae within 6 h, and spinosad + milbemycin oxime killed them within 7 h. Within 24 h, all fly larvae were killed with afoxolaner + milbemycin oxime. There was a synergism between spinosad and milbemycin oxime [101].

The rapid prevention of adult flea feeding has been shown to be important in reducing the effects of FAD. Fluralaner and afoxolaner both provided dramatic improvement of FAD in dogs [102]. Similarly, a topical application of fluralaner decreased all the FAD signs on cats beginning at day 7 and continuing until day 84 [56]. An oral dose of fluralaner resolved 90% of 20 cases of FAD at day 84 and 94% of 16 cases at day 168 [103]. FAD symptoms were dramatically reduced in dogs treated with sarolaner or spinosad at day 90 [70]. Oral applications of sarolaner or spinosad provided 62–67% reduction in FAD as measured by the Canine Atopic Dermatitis Extent and Severity Index-4 scale [71]. Spinetoram was better at reducing FAD compared with the fipronil + methoprene treatment [28]. The topical application of fluralaner provided significant reduction in clinical signs of FAD in cats [104]. Oral doses of sarolaner + moxidectin + pyrantel or afoxolaner + milbemycin oxime dramatically reduced the symptoms of FAD in naturally infested dogs within 30 days [48]. In a multiple location study in the USA, an oral dose of sarolaner + moxidectin + pyrantel or afoxolaner dramatically reduced the clinical signs of FAD with dogs [79].

### 5.4. Formulations

The use of pet collars has been popular for decades even though there was little published evidence that they controlled flea populations on pets [6]. However, collars containing 10% imidacloprid + 4.5% flumethrin provide effective control of fleas for up to 8 months [105,106,107,108]. In vitro lab studies have shown that the combination of imidacloprid and flumethrin synergizes their toxicity against fleas and ticks [109]. Collars containing imidacloprid + flumethrin provided a 100% reduction in flea numbers on dogs at days 120 and 210, whereas deltamethrin collars provided only 76.7 and 66.7% reduction at days 120 and 210, respectively [110]. The use of an imidacloprid + flumethrin collar on cats on the Isles of Lipari and Vulcano reduced flea infestations by 79.4, 100, and 93.6% at 210, 270, and 360 days, respectively [111].

In addition to controlling fleas and ticks, collars containing either imidacloprid + flumethrin or deltamethrin, provided 88.3 and 61.8% efficacy in preventing the transmission of *Leishmania infantum* by phlebotomine sand flies. In a study with cats, the imidacloprid + flumethrin collar prevented 75% of feline *Leishmania* infection [112]. In another study, imidacloprid + flumethrin collars provided a preventive effect of 71.5% in the transmission of *Bartonella* in cats [113]. The collar provides an effective measure to reduce the risks of both diseases.

## 6. Insecticide Resistance

A review of insecticide resistance indicates widespread resistance to certain carbamates, organophosphates, and pyrethroids [114]. More than 3000 *C. f. felis* populations were collected from 10 different countries from 2002 to 2017. Of the 1837 isolates that were tested, there was no evidence of a decreased susceptibility to imidacloprid [115]. In another study an isolate of *C. f. felis* collected from three dogs in which fipronil + methoprene spot-on had not performed well was reared and tested in the laboratory. The field-collected isolate was tested on cats and dogs treated with fipronil + methoprene. The fipronil + methoprene provided >97.6% reductions in fleas on dogs and >85.6% on cats for at least 29 days. The authors concluded that the reported failures in earlier trials were probably due to factors other than insecticidal resistance [116]. To date, there is little evidence suggesting that resistance has developed to many of the new topical and oral insecticides being used.

Populations of *C. f. felis* from goat farms in Turkey had >90% frequency of the *kdr* gene (L1014F) and super *kdr* gene (T929V) in response to intensive use of cypermethrin [117]. This study supports the claim that pyrethroid resistance is widespread in *C. f. felis* [112].

## 7. Control Strategies

Strategies to control cat fleas have evolved over the past two decades with the advent of new chemistries and therapies. Treatment of the pet, the indoor environment, and the outdoor environment was the standard practice prior to advent of modern topical and oral treatments in the mid-1990s [1,2]. In recent years, the reduction in the ability of cat fleas to lay viable eggs has become an important factor when considering the efficacy of topical and oral treatments of pets [118,119]. By interrupting flea reproduction, these topical and systemic flea products are capable of controlling flea populations in the indoor premises.

Of the 759 pet-owners surveyed from 84 different veterinary clinics from 2002 to 2012, 71% of the dog owners and 50% of the cat owners had used a flea control product in the previous 12 months. Pet owners preferred spot-on, on-animal sprays, and pills as treatment options. Some common causes of failures to control fleas were a lack of knowledge about the flea life cycle, misapplication of the product, and noncompliance of treatment schedules by the pet owners [120].

There has been a trend to market therapies having a combination of products that protect or control various ectoparasites and endoparasites. Products with active ingredients to control fleas and ticks include drugs like moxidectin (antihelminthic activity against heartworms) and pyrantel (anti-parasiticide against pinworms and roundworms). This might be in part explained by a recent survey of 24 veterinary hospitals across the country. About 96% of these veterinarians recommended 12-month flea and tick control. Of the 559 dog owners surveyed, 73% also felt 12 months control necessary. However, pet owners typically apply flea and tick control products for only 4 to 4.6 months [121]. When purchasing patterns of 650 veterinary clinics were examined from 2014 to 2017, some 201,565 dogs were prescribed either fluralaner (29.1%), afoxolaner (58.9%), or spinosad (26.9%). Approximately 80% of the dogs were from southern and Midwestern states. Dog owners purchased more months of protection by selecting longer duration products like fluralaner than shorter length treatments like afoxolaner or spinosad. However, the months of protection obtained by the dog owner was less than that recommended by a veterinarian by 53% for fluralaner, 62% for afoxolaner, and 71% for spinosad [122]. Purchasing products with longer lasting activity against endo- and ectoparasites is clearly a strategy that pet owners are likely to adopt.

Two distinctly different approaches to use of isoxazolines have developed. Pfister and Armstrong [123] provide a review and discussion of the merits of a cutaneous application (permethrin) and a systemic application (fluralaner). When they considered the following four factors, owner adherence to the recommended treatment protocol, rapid onset of activity following administration, uniform efficacy over all areas of the treated dog at risk for parasite attachment, and maintenance of high efficacy throughout the retreatment interval, they felt the systemic treatment provided the optimal outcome. If sarolaner or fluralaner had been chosen as the cutaneous treatment instead of permethrin, possibly their assessment might change.

There has been a reluctance in some regulatory agencies to consider oral or topical treatments of pets as a stand-alone treatment. However, evidence is accumulating that the treatment of all pets within a home, with many of the newer therapeutics, can break the lifecycle and completely control an indoor infestation [118]. A topical application of imidacloprid reduced the number of fleas on pets by 98.8% on pets and the number of fleas trapped in the homes by 99.9%. An oral dose of lufenuron combined with pyrethrin sprays reduced the number of fleas on animals by 99.2% and the number trapped in homes by 99.7% [31]. Three monthly topical applications of fipronil or imidacloprid to cats and dogs provided 96.5 and 99.5% reductions on animals and the number of fleas trapped within homes was reduced 98.6 and 98.6%, respectively [124]. In Australia, pets were treated with nitenpyram + lufenuron or imidacloprid. Nitenpyram + lufenuron provided 90–100% reduction of fleas on animals over the year and 100% reduction of fleas in the home [125]. Topical treatments with imidacloprid varied over the 1-year study. There was an 84.2–97.2% reduction of fleas on animals over 16 weeks. After the rainy season, control ranged from 70.5 to 87.8% reductions. The number of fleas trapped ranged from 0.0 to 93.6%. A single topical application of indoxacarb or monthly applications of fipronil + methoprene provided 99.1 and 54.8% reductions of fleas on dogs at day 60. Light trap counts in the homes decreased by 97.7% with indoxacarb and 84.6% with fipronil + methoprene at day 60 [126].

Similar results have been reported with isoxazoline compounds. Oral doses of afoxolaner for dogs reduced the number of fleas on dogs by 99.3 and 100% at days 7 and 30, respectively. The number of fleas caught in light traps in the residence decreased by 97.7 and 100% at 30 and 60 days, respectively [50]. Oral doses of fluralaner or afoxolaner provided 100% reduction in the number of fleas collected from treated dogs at day 86. Fluralaner and afoxolaner provided 100 and 98.9% reductions in the number of fleas caught in light traps in homes [102]. A single topical application of fipronil + pyriproxyfen on cats prevented flea egg development for 15 weeks in a simulated indoor environment [36]. Carpet disks from rooms with fipronil + pyriproxyfen treated cats prevented egg development. The treatment successfully interrupted the life cycle. Oral doses of sarolaner or spinosad to dogs resulted in >99.8% reduction in the number of fleas trapped within structures at 60 days [79]. Similarly, Dryden et al. [127] reported that a single topical application of fluralaner reduced the flea count on cats by 100% and the number of fleas in light traps by 99.9% by day 86.

## 8. Future Directions

The use of RNAi delivery systems to control insects of veterinary importance is an exciting new direction for the future of flea control. The delivery system allows for the knockdown of very specific targeted gene expression in both insects and acarines. Edwards et al. [128] were able to demonstrate the transfer of RNAi to cat fleas through a membrane feeding system, and resulted in 96% knockdown of GSTσ, a detoxification enzyme, within 2 days and sustained at least 7 days. The RNAi response to ingested dsRNA in *C. f. felis* was not impaired by gut enzymes of the flea. Another exciting delivery system that involves the use of nanoparticles is also being actively researched [129].

Even though there is little evidence to suggest widespread insecticide resistance in cat fleas to the modern arsenal of treatments available, it is important that flea populations be continually monitored to rapidly detect any changes in their populations. The maintenance of susceptible populations of *C. f. felis* is essential [112].

The control of *C. f. felis* in feral animals remains a problem. Feral animals provide for a reservoir of fleas in the environment. Currently, there is a lack of effective control measures to use outdoors to control these feral populations. Possibly baits to feed to feral animals containing actives such as the isoxazolines might be developed to control fleas, especially those associated with sylvatic plague.

## 9. Conclusions

The development of therapies that can reduce cat flea reproduction, prevent cat flea development, and rapidly kill adult fleas on the pet has dramatically altered our approaches to controlling cat fleas in the urban environment. The need for environmental treatments, especially indoors, has been greatly reduced. The costs of some of these new treatments may be prohibitive for universal adoption, but other adulticides containing IGRs are still effective and maybe more economical. The arsenal of potential topical and oral therapies to control cat fleas is impressive.

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
