# Peer review of "Recent Advancements in the Control of Cat Fleas"

_insects, 2020, doi:10.3390/insects11100668_

Round 1

Reviewer 1 Report

Dear author,

Your review is very clear. I made some comments on the pdf. I would like you to avoid speaking about pests and pesticides when dealing with animal parasites, even ectoparasites. It creates a confusion with agriculture and pests for which treatments do not get the same registration as veterinary medicine, not the same quality, not the same requirements, not the same formulations and dosages….

You did not define "biorational" so do not use it before clear definition as ou created this term.

Define natural and biological products, what is the difference?

Spinosyns must be a paragraph with chemicals, they are not naturals because extracted, purified and semi-synthesis like all macrocyclic lactones (slemacetin, ivermectin, moxidectin). They cannot be as "naturals" with essential oils.

All insecticidal molecules, groups are not "alternative" treatments!! they are the main treatments: 99% of the use by pet owners, > 2 billions dollars/year. What is alternative are plant extracts, essential oils….Change the title

You cannot mix paragraphs about chemical groups and paragraph about formulation like collars followed by chemicals. Collars contain chemical insecticides, and you only focus on imidacloprid collar, why? it needs to be clear that pyrethroids do not play a role against fleas but against ticks and as repellent. Only imidacloprid is active on cat flea.

Same for isox, they play no role against intestinal nematodes and heartworm, activity is due to moxidectin, selamectin or milbemycin  in the combination products. It is not clearly stated.

When you describe efficacy of new compounds: you must say if it was in vitro or on animal studies. It is really different . Fr some ) you must say how fleas were counted, at what time? for example for vaccine, were the counts 24h after infestation and how many times (monthly for one year?). To get a claim >95% of fleas must be killed in 24h.

I think it needs some moderate revision and rewritting.

Best regards

Author Response

I really appreciate the review and I have incorporated nearly all your suggestions. Attached is a note regarding the changes. 

Reviewer 2 Report

Comments to the manuscript insects-931918 intended as a review article in Insects “Alternative measures to control cat fleas” by MK Rust which gives an overview covering the new innovations and chemistries introduced the last two decades. The review updates the biorational approaches to control cat fleas.

This is an interesting, original, comprehensive and useful review; a solid contribution to the management of fleas and other arthropod parasites of dogs and cats.

Author Response

Thanks for the review.

Reviewer 3 Report

Dear Editor,

The article "Alternative Measures to Control Cat Fleas"  is an interesting and very well reviewed study. The author showed possible methods of the controlling of fleas using known veterinary medical products and IGRs, which influences the interruption of the flea life cycle. Moreover, a some drugs which have an impact on other parasites of cats and dogs were indicated.  

The described cases show different range of action of appropriate measures in reducing the occurrence of fleas in pets. Due to the many partial data in the text, the addition of a table grouping these data would be helpful in reading these messages, especially those methods/combinations of drugs that have proven to be the most effective and long-lasting in the control of fleas in  the pets. For this reason, I pointed to minor revision.

In my opinion the paper is more appropriate for journals of parasite control in pet.

Author Response

I had thought about a table, but I added some additional details about the studies instead. As I have indicated, the field trials with the new compounds have been very effective and I am not sure that anyone treatment is better.